# Potential Distribution and the Risks of *Bactericera cockerelli* and Its Associated Plant Pathogen *Candidatus* Liberibacter Solanacearum for Global Potato Production

**DOI:** 10.3390/insects11050298

**Published:** 2020-05-12

**Authors:** Jing Wan, Rui Wang, Yonglin Ren, Simon McKirdy

**Affiliations:** 1Harry Butler Institute, Murdoch University, Murdoch, WA 6150, Australia; 33386687@student.murdoch.edu.au (J.W.); Y.Ren@murdoch.edu.au (Y.R.); 2State Key Laboratory for Biology of Plant Diseases and Insect Pests, Institute of Plant Protection, Chinese Academy of Agricultural Sciences, Beijing 100193, China

**Keywords:** climate niche, early detection, insect–pathogen complex, invasive pests, landscape structure, potential distribution, risk assessment

## Abstract

The tomato potato psyllid (TPP), *Bactericera cockerelli*, is a psyllid native to North America that has recently invaded New Zealand and Australia. The potential for economic losses accompanying invasions of TPP and its associated bacterial plant pathogen *Candidatus* Liberibacter solanacearum (CLso), has caused much concern. Here, we employed ecological niche models to predict environments suitable for TPP/CLso on a global scale and then evaluated the extent to which global potato cultivation is at risk. In addition, at a finer scale the risk to the Australian potato acreage was evaluated. A total of 86 MaxEnt models were built using various combinations of settings and climatic predictors, and the best model based on model evaluation metrics was selected. Climatically suitable habitats were identified in Eurasia, Africa, South America, and Australasia. Intersecting the predicted suitability map with land use data showed that 79.06% of the global potato cultivation acreage, 96.14% of the potato production acreage in South America and Eurasia, and all the Australian potato cropping areas are at risk. The information generated by this study increases knowledge of the ecology of TPP/CLso and can be used by government agencies to make decisions about preventing the spread of TPP and CLso across the globe.

## 1. Introduction

In recent decades, there has been ever-increasing concern about biological invasions that pose large threats to food safety, biodiversity, and human activities [1,2]. Invasions of agricultural pests are particularly problematic because increasing global and regional trade of plant products can facilitate their introduction and spread [3,4]. Billions of dollars in economic losses have resulted from agricultural pest invasions worldwide [5,6].

The expanding distribution of psyllids globally over recent decades demonstrates how biological invasions have the potential to cause adverse impacts on natural and agricultural environments [7,8,9]. Psyllids (Hemiptera: Psylloidea), also called jumping plant lice, comprise many species that are important crop pests [9,10]. Psyllids damage plants both through feeding, which negatively affects plant growth, as well as acting as vectors of many plant pathogens [11,12,13,14]. Psyllids can be found in almost all regions of the world where solanaceous crop plants are grown, and in some regions, psyllid pests have caused severe economic losses with almost complete crop failure [10]. In addition to direct losses from crop failure and pest control costs, psyllids can also cause indirect losses such as a decline in agricultural exports due to biosecurity restrictions from importing countries [15].

There is much concern regarding the invasion and spread of the tomato potato psyllid (TPP), *Bactericera cockerelli* (Sulc) (Hemiptera: Triozidae), in Australasia, which includes Australia and New Zealand. TPP is native to Central and North America and has been identified as one of the most destructive solanaceous crop pests. In recent decades, TPP has been found to transmit the Gram-negative bacterium *Candidatus* Liberibacter solanacearum (CLso), which is a pathogen that results in severe yield and quality losses, primarily in potatoes and carrots [15,16,17]. It has been shown that the distribution of CLso in New Zealand and the Americas follows the dispersion of its psyllid vector, TPP [8]. CLso was reportedly introduced into New Zealand along with TPP from the western USA in the early 2000s through the horticulture trade. By the time CLso was first recorded in New Zealand, it had already spread to both the North and South Islands [17,18]. Introduction of TPP into new regions is likely to lead to the rapid spread of its associated plant pathogen CLso [17]. This indicates that the TPP and CLso insect–pathogen complex has enormous potential to expand toward other geographic regions of the world where habitats are favorable.

Tomato potato psyllid is a polyphagous insect that feeds on plants from more than 20 families, with a preference toward solanaceous crops (i.e., potato, tomato and eggplant) and solanaceous weeds (i.e., nightshade) [10]. In view of the wide availability of host plants, the risk of the global dispersion of TPP should be given priority consideration, particularly in regions where economically important crops such as potato and tomato are grown in large areas. The invasion and spread of TPP coupled with CLso may result in serious economic losses in these regions, and even endanger food security. For instance, the economic impact of TPP in the 4 years it has been in New Zealand is estimated in the millions of dollars in terms of increased management costs, crop losses and loss of export markets [6]. Additionally, there is growing concern about the environmental impact resulting from increased use of chemical pesticides [6]. For this reason, mapping the invasion risk areas to reveal the likely spatial variation of TPP and CLso and the potential consequences of invasion is imperative.

Ecological niche models (ENMs) are increasingly being applied to risk analysis of invasive pests because of their capacity to predict suitable habitats for pest colonization, allowing the adoption of biosecurity measures to prevent the invasion and spread of alien species in areas of concern [19,20]. Correlative models are the most commonly used method to predict the potential distribution of pests in a novel environment. This method connects species empirical observation data with bioclimate data to create a suitability gradient that can be projected onto a geographic space to generate a suitability map [20]. Additionally, a recent study revealed that psyllid population dynamics were strongly mediated by climate and landscape factors [21]. Here, ENMs coupled with spatial analysis were employed to investigate the potential risk of TPP and its associated plant bacterial pathogen CLso spreading around the globe. First, we compared the climate niche similarity between native and invasive populations of TPP and CLso using bioclimate data for their known sites of occurrence. Then, we employed correlative ENMs to forecast the suitable habitats available to TPP and CLso and produced a potential distribution map. Finally, by coupling ENMs with spatial analysis, we assessed the area of global potato cultivation and at a finer scale Australian potato production to determine the risk of establishment by the TPP and CLso insect–pathogen complex. Together, this information will be valuable for making decisions about how to prevent/address the invasion and spread of TPP/CLso to suitable regions. In particular this research focuses on the vegetable/potato planting areas of Australia.

## 2. Materials and Methods

MaxEnt (version 3.3.3k [22]) was selected to build the ecological niche models (ENMs) because it has been shown to be effective in predicting the potential distribution of invasive alien pests when utilizing present-only data [23,24]. Building models with proper complexity is crucial to prevent overfitting or underfitting, and to make robust inferences [25,26]. To build an optimal model for our target species, we optimized the following steps: (i) collecting and spatially filtering occurrence data; (ii) delimiting the background study area; (iii) Comparing the occupied climate space between native and invasive populations; (iv) selecting climate variables; and (v) configuring MaxEnt parameters (regularization multiplier, feature classes) and selecting the best model.

### 2.1. Occurrence Data Collection and Spatial Filtering

Occurrence records of TPP and CLso were collated from the literature, the Global Biodiversity Information Facility (GBIF, http://www.gbif.org/), the European and Mediterranean Plant Protection Organization Global Database (EPPO, https://gd.eppo.int/), and a report of occurrences in Australia (https://eldersrural.com.au/wp-content/uploads/sites/3/2017/03/) [27]. When only locality names were available, georeferenced coordinates were gained with the geolocation software Google Earth. To check and reduce spatial biases, these geo-referenced occurrence points were then subjected to spatial filtering to rarefy the points with a minimum distance of 50 km between each point [23,28]. This spatial filtering analysis was executed using SDMtoolbox [29] and resulted in 114 unique localities for TPP, of which 81 points were from the native regions in Central and North America and 33 points were from invaded regions in Australia and New Zealand. Similarly, 44 geo-referenced localities were collected for CLso, 21 of which were from the native areas in North America and 13 were from invaded regions in New Zealand.

### 2.2. Background Study Area Delimitation

MaxEnt, like other correlative ENMs, generates pseudo-absence points randomly sampled from the background area [23,30]. Previous studies indicated that background delimitation is a crucial step during the modelling process and can be achieved using different proxies [23,24,31]. Here, we selected the background study area by intersecting the occurrence localities with Koppen climatic zones downloaded from CliMond (http://www.climond.org) as this approach has been shown to be effective for other pests and is less arbitrary than defining a convex that encompasses all occurrence points [23,30,31,32,33,34]. The climatic zones with at least one occurrence record were selected as background (Figure 1). Random points were generated from the backgrounds to compare the climate niche similarity between invasive and native populations using SDMtoolbox [29].

### 2.3. Occupied Climate Space Comparison between Native and Invasive Populations

When a MaxEnt model is applied to predict the potential distribution of an alien species in a new range, the assumption that an alien species can maintain its climate niche in the invaded regions needs to be validated because the realized climate niche of alien species might shift during the invasion process [35,36,37]. Here, we carried out a principal component analysis (PCA) using the values of occurrence and random points extracted from 19 bioclimatic variables to analyze the climate niche similarity between native and invasive populations of TPP and CLso. A biplot was plotted with the first two components of PCA, and convex envelopes defining clusters of the invasive and native populations of TPP and CLso were added to visualize their climate niche overlap [34]. The bioclimatic variables were downloaded from the Worldclim database version 2.0 at a spatial resolution of 5 arcmin (http://www.worldclim.org) [38]. These Worldclim bioclimatic variables were employed to assess climate conditions because they include the climatic factors that determine species’ geographic distributions [39,40].

### 2.4. Climate Variable Selection

Previous studies have shown that climate variable selection is an important step for model fitting [41,42]. Here, two sets of variables were selected following the procedure suggested by Marchioro [23]. The first set of bioclimatic variables (Bio1, Bio2, Bio8, Bio12, and Bio15) was selected based on previous distribution modeling and life cycle adaption studies of other psyllid species [9]. The second set of bioclimatic variables was determined by adding the Bio14 variable to the first set according to PCA. We also calculated Pearson’s correlation coefficients using ENMtools software [43] to make sure that there was no multicollinearity between the selected variables [44].

### 2.5. MaxEnt Parameter Configuration and Best Model Selection

Recent studies have shown that using the default automatic configuration of MaxEnt may not always be appropriate [23,26,31]. It is recommended that the most appropriate model should be selected by evaluating the best potential combination of parameters (regularization multiplier, feature classes) [25,45,46,47]. Thus, we compared models with different feature class and regularization multiplier combinations. MaxEnt includes five basic feature classes: Hinge (H), linear (L), product (P), quadratic (Q), and threshold (T). As simple models with great explanatory predictive power can potentially be produced using various combinations of the feature classes [45,46], seven combinations were tested: L, H, LQ, LQP, LQH, LQPT, and LQHPT. The regularization multiplier values were set to 0.5, 1 (default), 3, 5, 7, and 9 based on Marchioro [23], Kumar et al. [46] and Morales et al. [47]. Combining regularization multipliers and feature classes, we assessed a total of 86 models for two environmental datasets, including two default auto-feature models.

Both threshold-dependent and threshold-independent metrics were employed to evaluate model performance. The threshold-independent metrics were the area under the curve (AUC) in a receiver operating characteristic (ROC) plot and the Bayesian Information Criterion (BIC). An AUC value of 1.0 indicates perfect discrimination ability and a value of 0.5 or less indicates a prediction no better than random [20]. The BIC criterion for model selection measures the trade-off between model fit and complexity, and the model with the lowest BIC is preferred [25,48]. The software ENMtools V1.3 was employed to calculate BIC [43].

Threshold-dependent metrics were the omission rate (OR) at the minimum training presence threshold (MTP) and OR at the 10% training presence threshold (TP10). The expected OR value is 0.1 at the TP10 and 0 at the MTP. Values higher than expected indicate the performance of the model is poor [28,49]. The following criteria were adopted to select the best model with low complexity and high performance: Lower BIC values, OR at TP10 and MTP approximate to 0.1 and 0, respectively, and higher AUC values (>0.8).

### 2.6. Model Projection to Predict the Potential Distribution of TPP and CLso

Once the parameter combination yielding the best model was determined, the MaxEnt model was run with all the known occurrences from native and invaded areas and projected onto the remaining parts of the world to predict the potential distribution of TPP and CLso. However, interpretation of model predictions outside the range of the independent variables on which models were calibrated is problematic [50]. A multivariate environmental similarity surface (MESS) implemented in MaxEnt was computed to quantify the extent of the environmental differences between model training and model projection data [44,51]. To increase the accuracy and reliability of modeling results, the final model was run for 30 replications and output in logistic format. Binary maps showing unsuitable, suitable and optimal habitats for TPP and CLso were then produced using the thresholds MTP and TP10. Habitats with logistic output values less than the MTP were regarded as unsuitable. In a similar way, habitats with values above the MTP and TP10 were considered suitable, and optimal respectively.

### 2.7. Spatial Analyses for Quantifying the Area at Risk of Attack

In addition to climate suitability, a recent study indicated that the landscape structure (i.e., host availability) and their spatial arrangement of the host can also determine the occurrence and abundance of pests and thus affect the damage to invaded ecosystems [21]. Here, we further integrated landscape pattern with climate suitability to quantify the area at risk of attack. According to previous studies, TPP and CLso primarily feed on potatoes, tomatoes and capsicums, but can be found on approximately 20 other plant families [10,16,17]. As potato is the third most important food crop worldwide, we first quantified the global potato production area at risk of attack by intersecting the TPP suitability map and the global potato distribution map. The global potato production area was obtained from geo-referenced data of potato-producing areas [52,53]. The acreage at potential risk of attack was calculated using SDMtoolbox with ArcGIS [29].

Next, we quantified the area at risk for potential TPP invasion in recently invaded areas of Western Australia by overlapping the TPP suitability map and a national scale land use map of Australia (https://data.gov.au/data/dataset/land-use-of-australia-2010-11). In addition to the cropping and horticulture areas, the residential and farm infrastructure, production forests, and modified grazing pastures were recognized as potential risk areas with available hosts such as backyard tomatoes and solanaceous weeds where TPP is likely to be introduced by unintentional human activities. This is because the new occurrences of TPP in Australia were mainly found in backyards containing tomatoes and eggplants [27]. Previous research also showed that non-crop host plants adjacent to cropping areas are important in the life cycle and ecology of TPP and CLso; this is because the insect’s life stages are present year-round and these host plants provide suitable feeding and breeding substrates throughout the year [16]. Similarly, natural conservation areas far from cropping areas can be recognized as risk areas but with low potential of invasion.

## 3. Results

### 3.1. Occupied Climate Space Comparison between Native and Introduced Populations

TPP was found across nine and three Koppen climate zones in its native America and invaded regions in Australasia, respectively. TPP occurred in various climatic zones from tropical to temperate in native regions and only occurred in warm and temperate climatic regions in invaded regions (Figure 1). Defining the occupied climate space by PCA allowed us to investigate niche similarity and divergence. The first two principal components of the PCA captured 72.4% of the total variation and these two components were significant. A high degree of overlap between the niches of native and introduced populations of TPP and CLso was observed (Figure 2). The available climate spaces in the native and invaded regions form two overlapped clouds, indicating that the available climate space in Australasia is only a part of the occupied climate space in its native habitat in America.

### 3.2. Model Calibration and Evaluation

Overall, 86 MaxEnt candidate models built with various combinations of regularization multiplier, feature class and climatic variables were evaluated to select the best fitting model to predict the potential distribution of TPP and CLso (Figure 3). Both threshold-independent (AUC, BIC) and threshold-dependent (MTP, TP10) evaluation metrics used to assess model performance varied with different parameter combinations. Some models showed ORs close to the expected values, whereas others showed ORs of up to 0.26, almost three times the expected value. AUC values ranging from 0.74 to 0.82 indicated that all models performed better than random. All evaluation metrics changed with different regularization multipliers. The change in the evaluation metrics was nonlinear and generally consistent between the four metrics. The lowest ORs and BIC values and highest AUC values were obtained when the regularization multiplier was 3. Similar trends were seen for different feature classes. The models built with the LQ feature usually had lower ORs and BIC values and higher AUC values. Although the variation in evaluation metrics was consistent between the two climatic sets, the values of evaluation metrics for models built with climatic variables set 2 were subtly higher or lower than those for models built with climate variables set 1. Based on the model selection criteria, the best model was obtained when using L and Q features, a regularization multiplier equal to 3, and climatic variables set 2 (Bio1, Bio2, Bio8, Bio12, Bio14, Bio15); this model had the lowest OR and BIC values, as well as an AUC more than 0.8. The performance of the selected best model was better than that of the MaxEnt model obtained using the default settings (Figure 3).

### 3.3. Potential Global Distributions of TPP and the Bacterial Pathogen It Transmits

Predicted climatic suitability maps with logistic and binary outputs are shown in Figure 4. The suitable and optimal areas were mainly distributed between 47° S and 65° N. In addition to the known regions in Central and North America, four vast climatically suitable and optimal regions were identified in South America, Eurasia and North Africa, sub-Saharan Africa, and Australasia. The optimal areas in South America were in the Andean Highlands and Pampas. The largest optimal area was in Eurasia and North Africa, and largely consisted of regions around the Mediterranean and a belt running from northwestern to southern China and continuing into the Gangetic plains in northern India and Bangladesh. Botswana, Zimbabwe, Southern Africa, Southern Australia, and most parts of New Zealand were also climatically optimal regions.

MESS analysis identified environments that exist in the model’s calibrated regions but not in the model’s projection areas, and these non-analog environments are shown in Figure 4b. These areas included Mauritania, Mali, Niger, Chad, Sudan, and Southern Algeria in Africa, the Tibet Plateau region in Asia and most regions above 60° N latitude in Europe and North America.

### 3.4. Risks to Global Potato Production and Australian Crop Production

The predicted suitable and optimal areas for TPP and CLso almost completely overlap with the global potato cultivation area; 79.06% of the known global potato cultivation acreage and 96.14% of main potato production acreage in South America and Eurasia were predicted as both suitable and optimal areas for TPP and CLso (Appendix A). The newly invaded areas that are at high risk for potential invasion are located in eastern, western and southern Australia, and include different land use types with host availability. The acreage of Australian lands under risk of attack varies widely between land use types (Figure 5). The cropping and horticulture areas are at highest risk, with almost all the area within the optimal range for TPP and CLso, followed by residential, transport and communication areas (97.3%), plantation forest and grazing modified pasture (88.0%), and nature conservation areas (38.67%). The known sites of occurrence in Australia were mainly located in residential regions surrounded by cropping and horticulture areas (Figure 5).

## 4. Discussion

More and more invasive alien pests are being recognized as having an adverse effect on crop production, biodiversity, economies and society [2,4]. Quantitative assessment or prediction of the probability of an alien pest invasion and creation of a risk map conveying the spatial variation of a pest is the key to developing strategic and tactical approaches for invasive pest management [19]. Frequently, predicting an invasion is dependent on prediction of climate suitability using extrapolations made from limited information to project how a species might arrive, establish, or spread in novel environments and impact these environments [19,20]. In particular, a recent study found that the occurrence and abundance of TPP in its native habitat in the USA could be best described by incorporating climate and landscape factors [21]. Here, we applied MaxEnt models with known occurrence data and spatial bioclimatic layers to predict areas climatically suitable for establishment of the TPP/CLso complex on a global scale and then combined information about climate suitability from these models with spatial land use layers to assess the risks of invasion in global potato cultivation areas and major crop production regions in Australia that have recently been invaded.

According to the ecological niche model assumption, we evaluated niche conservatism before model calibration [20,35,39]. No niche shift was found between native and invasive populations of TPP and its associated pathogen CLso. The occupied niches of CLso were found within those of its host TPP and this niche similarity provided us a chance to predict the potential distribution of TPP and CLso as a complex, as it is usually difficult to detect the pathogen [8,21]. Niche comparison further indicated that the climate space occupied by the invasive populations is only a portion of that occupied in their native regions, implying that TPP and CLso may continue to expand their range in Australasia unless efficient biosecurity measures are taken.

The performance of 86 candidate models varied largely with changes to MaxEnt’s settings. Values of the regularization multiplier had the most impact on model performance, followed by combinations of feature class and climatic variables. Nonlinear variation of model performance with different regularization multiplier values and combinations of feature classes revealed that an appropriate degree of complexity is an ideal property for improving the transferability of ENM models from native to non-native regions when using an ENM model to predict potential distributions, as previous studies indicated [45,48,54,55]. Therefore, our results corroborate the findings of other studies [23,25,26,47,48] that it is important to build a MaxEnt model for specific species by testing different combinations of parameters instead of adopting default settings, and that the optimization model should have an appropriate level of complexity.

The final selected climatic suitability model for the TPP/CLso complex revealed four large regions suitable for invasion and establishment in South America, Eurasia, Africa, and Australasia. However, we cannot absolutely infer that TPP and CLso cannot survive in the unsuitable areas because there are some limitations to our predicted potential invasion areas. Our MaxEnt model, built with occurrence data, predicted the realized niche, which is regulated by both biotic interactions and abiotic factors that shape the species distribution [20,35,39]. Potentially important biotic interactions (competition with local species, presence/absence of natural enemies, population recruitment) were not taken into account due to the lack of relevant information for most psyllid species. Acquiring such information is hindered by the fact that psyllids are small insects and often overlooked in general biodiversity collecting [9]. Our model projection outside of the native range is thus a relative approximation of the climate niche. It is possible that TPP and CLso could survive in the areas that were predicted as being of low suitability when the amount of the TPP/CLso complex introduced from occurrence regions is high enough. But despite the above drawbacks, the model we built is valuable and informative and provides a fundamental tool for predicting suitable areas for the TPP/CLso complex, revealing areas that are more vulnerable to invasion and establishment than those with unsuitable conditions. Large potentially suitable areas outside its native range suggest that TPP and its associated plant pathogen CLso should be considered an emerging global crop/pest complex.

According to our analysis, substantial portions of Eurasia, South and North Africa, South America, and Australasia were identified as climatically suitable areas with hosts available for TPP and CLso. Most of the host plants of TPP in its native America, including cultivated and wild Solanaceae species, are widely distributed in the predicted suitable regions and may form a plant corridor that promotes the invasion and spread of TPP and CLso. The invasion and spread of this complex to the predicted suitable and optimal regions may cause significant economic losses for local crop producers, because almost all the acreage cultivated with potato, the third most important food crop worldwide, is located within these regions. The use of pesticides to control TPP might also have potential impacts on the local environment and thus increase social costs. It is imperative to formulate biosecurity measures to prevent the global invasion and spread of the TTP/CLso complex, particularly in the newly invaded regions of Australia. Strict quarantine measures, particularly for crop and horticulture products from regions with known occurrence of TPP and CLso, should be adopted for the countries and regions identified as high risk areas with a suitable climate and hosts available for TPP and CLso.

Although prevention strategies before pests have established viable populations in a novel region is broadly considered more cost-efficient than eradication or control of the invading populations [56,57], not all prevention methods are effective. This is particularly true in the current globalization era with increasing global and regional communication; even the best prevention efforts cannot stop all invasions of alien species. Early detection coupled with rapid response is a critical second defense against the establishment of newly invaded populations. TPP has already been introduced in New Zealand and Australia, and it is thought to be in the earlier stages of invasion in these regions, which highlights the importance of early detection and rapid response to increase the likelihood that localized newly invaded populations will be found, contained, and eradicated before they become widely established. To increase the probability of detecting established populations, it is important to identify highly vulnerable regions with suitable climates and host plants available for the survival of introduced propagules. Our spatial analysis performed by overlaying suitable areas and land use types delimited the areas at risk for potential invasion in Australia. Nearly all the crop areas and residential areas in Eastern, Southern, and Western Australia are located in the optimal climate regions and thus can be recognized as high-risk areas.

Considering that the currently invaded localities are mainly confined to non-cropping areas (i.e., backyards of residential regions) and that a related study revealed the TPP in its native habitat in the USA was more abundant in landscapes with high connectivity, low crop diversity and large natural areas [21], monitoring efforts should put more emphasis on the corridors or routes connecting the currently invaded localities and the neighboring crop cultivation regions, particularly those planted with potato and other Solanaceae mono-crops. In summary, the risk maps generated here can be used by biosecurity policy makers and frontline practitioners to delimit priority areas for installing detection traps and conducting field surveys, and to coordinate management efforts strategically and tactically in areas at risk of invasion so as to prevent the invasion and spread of TPP beyond the currently occupied areas as soon as possible.

## 5. Conclusions

Our study highlights the importance of integrating climate and landscape factors using ENM and spatial approaches to identify the areas at risk from invasive pests. Species-specific ENMs should be built with appropriate complexity by configuring the potential parameters to characterize the climate niche and to predict the outbreak of pests across variable landscapes. Large climatically suitable regions with available hosts were identified in Eurasia, South and North Africa, South America, and Australasia. Spatial analysis indicated that predicted suitable areas highly overlap with global potato cultivation areas: 96.14% of the main potato production acreage in South America and Eurasia, and all the Australian potato growing areas are under potential risk of invasion. Our results and generated risk map can provide scientific guidance for implementing early detection or eradication measures and thus prevent the introduction or spread of TPP and CLso over the globe. In addition, our study contributes to the ecological knowledge of TPP and CLso, and could serve as a guide for further experiments to develop novel models for assessing the potential invasion and impact of this pest/pathogen complex.

## Figures and Tables

**Figure 1 insects-11-00298-f001:**
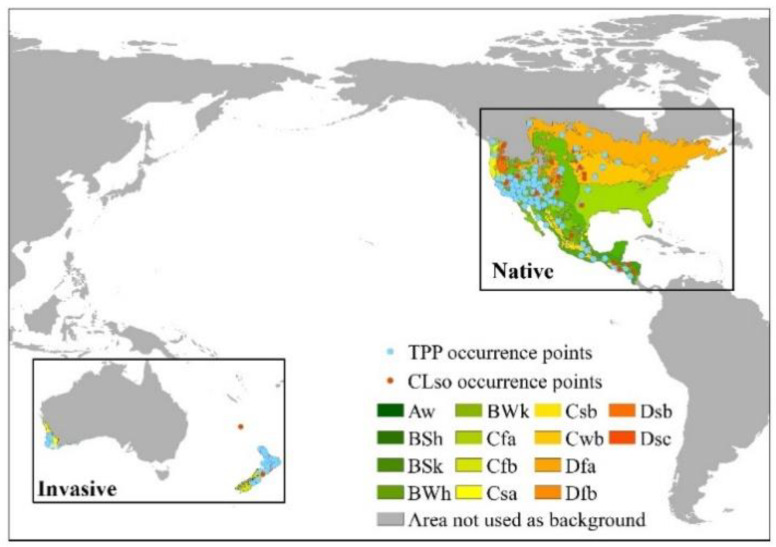
Occurrence points and delimited background for native and invasive populations of tomato potato psyllid (TPP) and *Candidatus* Liberibacter solanacearum (CLso). Colors refer to the Koppen climate zones, and gray represents non-target background. Letter codes refer to climate classification: **A**, equatorial; **B**, arid; **C**, warm temperate; **D**, snow; **W**, desert; **S**, steppe; **a**, hot summer; **b**, warm summer; **c**, cool summer; **f**, fully humid; **h**, hot arid; **s**, summer dry; **w**, winter dry.

**Figure 2 insects-11-00298-f002:**
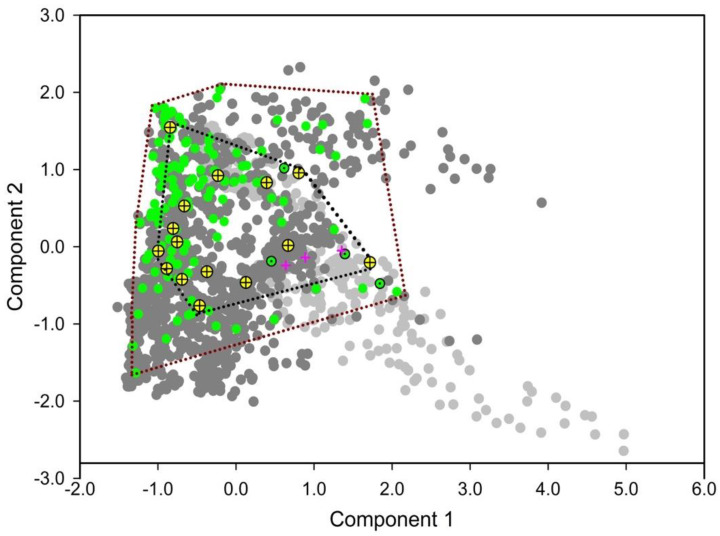
Comparison of climatic niches between native and introduced populations using principle component analysis. Green and green dotted circles represent native and invasive populations of tomato potato psyllid (TPP), respectively. Yellow crossed circles and pink plus symbols are native and invasive populations of *Candidatus* Liberibacter solanacearum (CLso), respectively. Light and dark gray dots depict random points generated from invaded and native backgrounds, respectively.

**Figure 3 insects-11-00298-f003:**
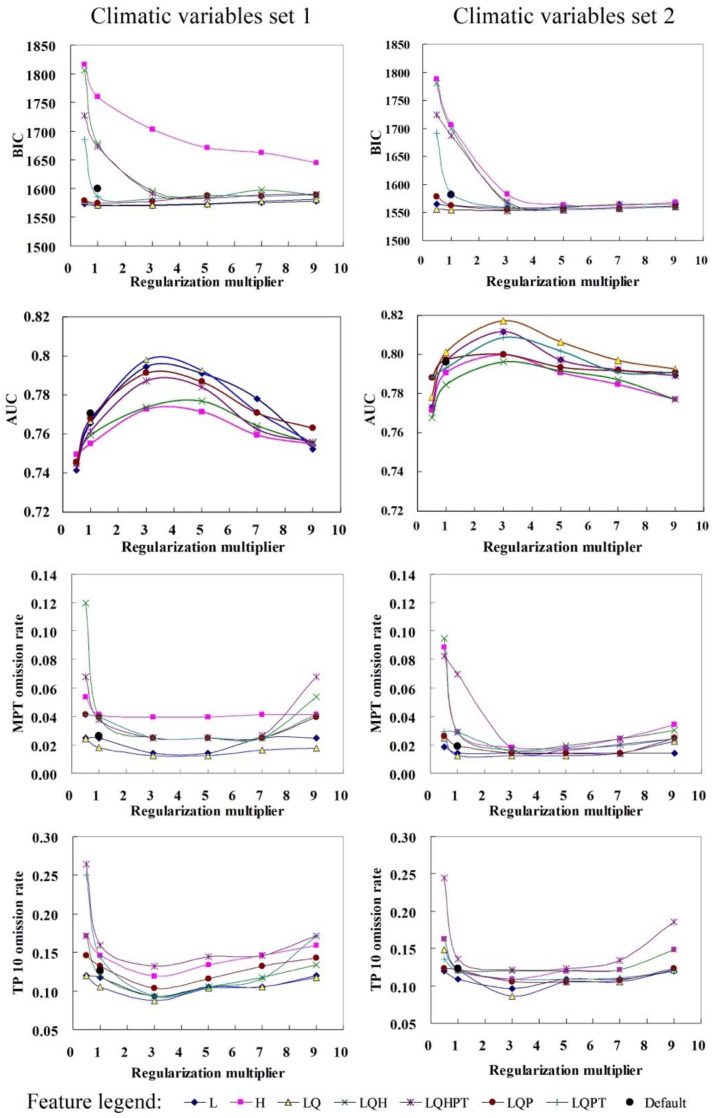
Performance statistics for models of tomato potato psyllid (TPP) distribution built with various combinations of regularization multiplier, feature class and climatic variables. (Feature abbreviations: **L**, linear; **Q**, quadratic; **P**, product; **H**, hinge; **T**, threshold).

**Figure 4 insects-11-00298-f004:**
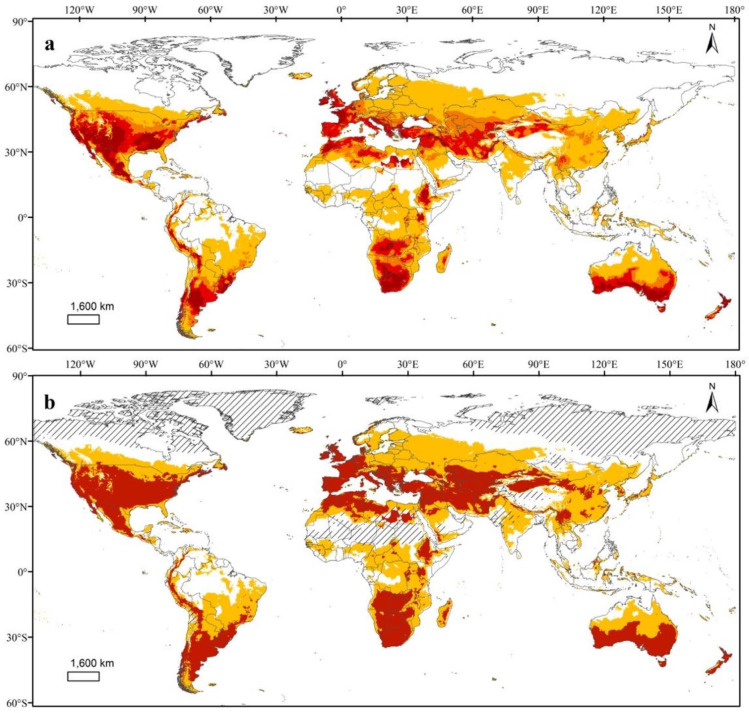
Predicted suitable habitats for tomato potato psyllid (TPP) and the associated plant pathogen *Candidatus* Liberibacter solanacearum (CLso) shown as logistic (**a**) and binary (**b**) output. In the logistic map, dark red colors represent higher suitability. Orange and red colors in the binary map represent suitable and optimal conditions for TPP and CLso, defined by the minimum training presence threshold (MTP) and 10% training presence threshold (TP10), respectively. The black simple hatch lines in the binary map indicate the non-analogous environments between the model’s calibration and projection areas identified by MESS analysis.

**Figure 5 insects-11-00298-f005:**
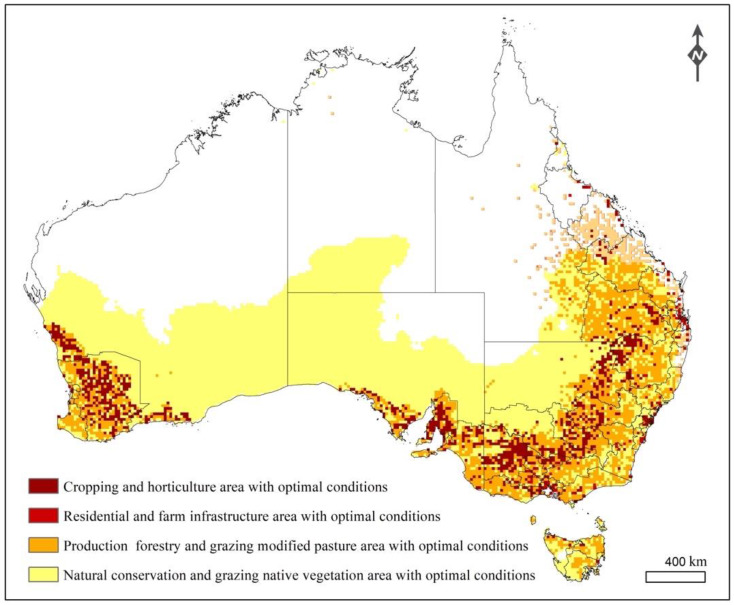
Potential areas in Australia at risk for invasion identified by spatial overlay analysis of predicted climate suitability for tomato potato psyllid (TPP) and *Candidatus* Liberibacter solanacearum (CLso) and national scale land use data.

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
