# Peer review of "Potential Distribution and the Risks of Bactericera cockerelli and Its Associated Plant Pathogen Candidatus Liberibacter Solanacearum for Global Potato Production"

_insects, 2020, doi:10.3390/insects11050298_

Round 1

Reviewer 1 Report

Generally, this paper well written and the results are clearly presented. The statistics is appropriate. Nonetheless, there are some corrections that should be revised before publication.

The important reference paper, Illan et al (2020), is not mentioned in this MS. I attached the PDF file. Some results and discussion parts seem similar meaning to your study. The authors should compare/discuss the results from your MS and them. I expect the discussion section will be revised, so I suggested "Major Revision".

L12 to his work -> to this work?

L112-115: This part can be shorter.

L131-137: This part also can be shorter.

L414-416: Please add the “Year”

Author Response

Reviewer’s comments:The important reference paper, Illan et al (2020), is not mentioned in this MS. I attached the PDF file. Some results and discussion parts seem similar meaning to your study. The authors should compare/discuss the results from your MS and them. I expect the discussion section will be revised, so I suggested "Major Revision".

Author’s reply: Many thanks for you for providing us with the newly published paper by Illan et al., 2020, which is focused on characterizing factors affecting potato psyllid (i.e. tomato potato psyllid, TPP in our study) occurrence and abundance in its native habitat in the USA. This study used regional multi-year monitoring data (2012-2017) collected from a region encompassing 401460 km2 of the states of Idaho, Oregon and Washington to assess landscape and climate drivers of TPP populations. Their results show that populations of TPP were strongly affected by landscape and climate factors. TPP were more abundant in landscapes with high connectivity, low crop diversity and large natural areas. This study indicates that climate and landscape factors can effectively describe the population dynamics of TPP in an intensively managed ecosystem and thus provides us with strong justification for integrating climate suitability and host availability to assess the potential risk posed by TPP and its associated pathogen CLso to global and regional potato production. As the reviewer noted, some parts of the results and discussion of Illan et al. (2020) seem similar to our study. In terms of methodology, the study of Illan et al. (2020) and ours are similar in that climate and landscape factors were incorporated by applying SDM/ENM to describe the occurrence and abundance of TPP. However, in terms of aims or questions, our study is different from Illan et al. (2020). Our study aimed to infer the climate gradient suitable for TPP using information from its known sites of occurrence in its native habitats and then projecting that information onto other parts of world. By contrast, Illan et al (2020) aimed to characterize factors affecting TPP occurrence and abundance at a regional scale in the USA. In this context, we made a minor revision in the introduction, methods and results sections. In addition, their finding that landscape connectivity between crops can affect the occurrence and abundance of TPP provides us with valuable insight into the importance of establishing a monitoring network in the newly invaded areas of Australia, as the habitats at risk such as cropping and horticulture, residential and farm and natural areas overlap and this highly connected landscape structure might promote the establishment of TPP populations in the backyards of residential areas of Australia, allowing TPP to spread rapidly towards cropping and horticulture areas and thus threatening potato and vegetable production. In this context, we have added suggestions on how to establish a monitoring network to prevent expansion from established populations in the discussion section (Lines 387-392 in the revised version).

In summary, major revisions were made to the discussion and conclusion sections, and minor revisions were made to the introduction, methods and results sections. Illan et al. (2020) was added to the reference list, and the reference numbers were changed accordingly.

Reviewer’s comments:L12 to his work -> to this work?

Author’s reply: Revised.

Reviewer’s comments:L112-115: This part can be shorter.

Author’s reply:

Lines 113-119 in original manuscript, reworded sentence to “Previous studies indicated that background delimitation is a crucial step during the modelling process and can be achieved using different proxies [30,22,23].”

Lines 119-120 in original manuscript, reworded sentence  to “Here, we selected the background study area by intersecting the occurrence localities with Koppen climatic zones downloaded from CliMond (http://www.climond.org) as this approach has been shown to be effective for other pests and is less arbitrary than defining a convex that encompasses all occurrence points [22,29,33].”

Reviewer’s comments:L131-137: This part also can be shorter.

Author’s reply:

Lines 131-133 in original manuscript, deleted sentences, “MaxEnt is a correlative niche-based model that has been commonly employed to predict the presence or absence of species in unsampled locations by relating species observations to climatic variables.”

Lines 136-137 in original manuscript, deleted sentence “If the climatic niche is maintained, the MaxEnt model calibrated in the native region can be projected onto the target region to predict the potential invasive distribution.”

Reviewer’s comments:L414-416: Please add the “Year”

Author’s reply: This reference is replaced by a new one from same author. Munyaneza, J. E. Zebra chip disease of potato: biology, epidemiology, and management. Am J Potato Res 2012, 89, 329–350.

Reviewer 2 Report

The manuscript gives a good contribution to investigate the potential risk of Bactericera cockerelli and its associated plant bacterial pathogen (bacterium Candidatus Liberibacter solanacearum) spreading over the globe. The study is focused, relevant and novel; also, the manuscript is well-written. Provided they conduct changes to the manuscript, I believe this paper could be of interest to the interested reader on ecological models of agricultural pests.
A few points:
L.18: …acreage are at risk of…
Ls.28-29: Keywords should be in alphabetic order. Also, keywords serve to widen the opportunity to be retrieved from a database. To put words that already are into title and abstracts makes KW not useful. Please choose terms that are neither in the title nor in abstract.
L.33: Place “,” after biodiversity
Ls.91-92: Place by
Ls.91-93: Delete this sentence
Ls.113-119: Summarize these sentences
L.163: Place “,” after LQPT
L.296: bioclimatic?
L.338: Delete “,”
L.348: Although prevention strategies…
L.362: …as high-risk areas.

Author Response

Comments and Suggestions for Authors

The manuscript gives a good contribution to investigate the potential risk of Bactericera cockerelli and its associated plant bacterial pathogen (bacterium Candidatus Liberibacter solanacearum) spreading over the globe. The study is focused, relevant and novel; also, the manuscript is well-written. Provided they conduct changes to the manuscript, I believe this paper could be of interest to the interested reader on ecological models of agricultural pests. A few points:

Reviewer’s comments:L.18: …acreage are at risk of…

Author’s reply:

This sentence was replaced by a new one in the revised version (Line 19).

Reviewer’s comments:Ls.28-29: Keywords should be in alphabetic order. Also, keywords serve to widen the opportunity to be retrieved from a database. To put words that already are into title and abstracts makes KW not useful. Please choose terms that are neither in the title nor in abstract.

Author’s reply: Deleted the words found in the abstract and title (Bactericera cockerelli, Candidatus Liberibacter solanacearum (CLso), ecological niche modelling), and added new words.

The revised keywords listed in alphabetic order are climate niche; early detection; invasive pest management; insect-pathogen complex; landscape structure; potential distribution; risk assessment.

Reviewer’s comments:L.33: Place “,” after biodiversity

Author’s reply: Revised.

Reviewer’s comments:Ls.91-92: Place by; Ls.91-93: Delete this sentence

Author’s reply:  

Lines 91-93 in original manuscript, modified sentence to “Building models with proper complexity is crucial to prevent overfitting or underfitting, and to make robust inferences [24,25].”

Lines 93-94 in original manuscript, reworded sentence to “To build an optimal model for our target species, we optimized the following steps:”

Reviewer’s comments:Ls.113-119: Summarize these sentences

Author’s reply:

Lines 113-119 in original manuscript, modified sentences to “Previous studies indicated that background delimitation is a crucial step during the modelling process and can be achieved using different proxies [30,22,23].”

Reviewer’s comments:L.163: Place “,” after LQPT

Author’s reply: Changed.

Reviewer’s comments:L.296: bioclimatic?

Author’s reply: Changed.

Reviewer’s comments:L.338: Delete “,”

Author’s reply: Deleted.

Reviewer’s comments:L.348: Although prevention strategies…

Author’s reply: Revised.

Reviewer’s comments:L.362: …as high-risk areas.

Author’s reply: Revised.

Round 2

Reviewer 1 Report

The MS was well revised.

This manuscript is a resubmission of an earlier submission. The following is a list of the peer review reports and author responses from that submission.